# Anatomy-Guided Multi-Path CycleGAN for Lung CT Kernel Harmonization

**Aravind R. Krishnan**[1]                                              ARAVIND.R.KRISHNAN@VANDERBILT.EDU

**Thomas Z. Li**[1]                                                          THOMAS.Z.LI@VANDERBILT.EDU

**Lucas W. Remedios**[1]                                          LUCAS.W.REMEDIOS@VANDERBILT.EDU

**Kaiwen Xu**[2]                                                                   KAIWEN.XU@INSITRO.COM

**Lianrui Zuo**[1]                                                          LIANRUI.ZUO@VANDERBILT.EDU

**Kim L. Sandler**[3]                                                             KIM.SANDLER@VUMC.ORG

**Fabien Maldonado**[3]                                                 FABIEN.MALDONADO@VUMC.ORG

**Bennett A. Landman**[1,3]                                     BENNETT.LANDMAN@VANDERBILT.EDU

[1] *Vanderbilt University, Nashville, TN, USA*

[2] *Insitro, South San Francisco, CA, USA*

[3] *Vanderbilt University Medical Center, Nashville, TN, USA*

**Editors:** Accepted for publication at MIDL 2025

## Abstract

Accurate quantitative measurement in lung computed tomography (CT) imaging often relies on consistent kernel reconstruction across scanners and manufacturers. Harmonization can reduce measurement variability caused by heterogeneous reconstruction kernels; however, harmonization across different manufacturers and scanners remains challenging due to significant differences in reconstruction protocol and positional alignment of subjects, often resulting in anatomical hallucinations. To address this, we propose a multi-path cycleGAN framework that incorporates multi-region anatomical labels and a tissue statistic loss as anatomical regularization to preserve structural integrity during harmonization. We trained our model on 100 scans each of four representative reconstruction kernels from the National Lung Screening Trial (NLST) dataset and evaluated it on 240 withheld scans. Experimental results demonstrate superior performance of our method in both within-manufacturer harmonization and cross-manufacture harmonization: Harmonizing hard-to soft-kernel images within a single manufacturer significantly reduces emphysema measurement discrepancies ($p < 0.05$). Across manufacturers, harmonizing all kernels to a reference soft kernel yields consistent emphysema quantification ($p > 0.05$) and preserves anatomical structures, as demonstrated by improved Dice similarity coefficient in skeletal muscle and subcutaneous adipose tissue between harmonized and unharmonized images. These findings demonstrate that segmentation-driven anatomical regularization effectively addresses cross-manufacturer discrepancies, ensuring robust quantitative imaging. We release our code and model at https://github.com/MASILab/AnatomyconstrainedMultipathGAN.

**Keywords:** cycleGAN, harmonization, CT, emphysema, synthesis

## 1. Introduction

The reconstruction kernel in computed tomography (CT) is a scanner parameter that impacts the spatial resolution and signal to noise ratio (SNR) of the image. The choice of

kernel introduces a trade-off between spatial resolution and noise, where images reconstructed with a "hard" kernel have high spatial resolution but poor SNR. On the other hand, images reconstructed with a "soft" kernel have decreased spatial resolution but improved SNR (Schaller et al., 2003). Hard kernels highlight structures such as bone and lung, while soft kernels are useful for soft tissues (Lasek and Piórkowski, 2020). However, the acquisition protocol varies across manufacturers, causing undesired variability in "sharpness" and "softness" of different kernels. This flexibility of reconstruction kernels often introduces inconsistencies in quantitative imaging measurements, including percent emphysema (Boedeker et al., 2004), body composition assessment (Troschel et al., 2020), radiomic feature assessment (Meyer et al., 2019), and coronary artery calcification (Lessmann et al., 2017). This issue is further compounded in multi-center and longitudinal studies, where acquiring CT images with consistent reconstruction kernels over time is challenging. To mitigate this issue, CT harmonization techniques have been proposed in recent years (Krishnan et al., 2024a,b).

Cross-vendor harmonization is beneficial in multi-center studies and longitudinal studies where paired data might be unavailable. However, cross-vendor harmonization is challenging due to variability in reconstruction protocols and positional alignment of subjects. In recent times, the cycleGAN (Zhu et al., 2017) has been adapted for cross-vendor harmonization (Gravina et al., 2022; Yang et al., 2021; Selim et al., 2022). Our group (Krishnan et al., 2024a) introduced a multipath cycleGAN model with a shared latent space that could minimize differences in emphysema measurements across paired and unpaired kernels. However, the difference in the semantic field of view (FOV) across the images resulted in artifacts outside the FOV. To address these challenges, we modified the shared latent space, standardized preprocessing by enforcing circular FOVs, and developed a two-stage multipath cycleGAN model (Krishnan et al., 2025), hypothesizing that a shared latent space enforces consistency in emphysema measurements and preserves anatomy post harmonization. Additionally, we incorporated an identity loss to preserve radio-opacity post-harmonization. Stage one focused on harmonizing different kernel combinations from Siemens and GE scanners, while in Stage two, the pre-trained kernels were frozen and kernels from Philips and GE were harmonized across all combinations of kernels from both stages.

The modified multipath cycleGAN achieved consistent emphysema measurements across paired and unpaired kernels. However, when the GE BONE kernels were harmonized to the reference Siemens B30f kernel, certain regions of the skeletal muscle and subcutaneous adipose tissue (SAT) intensities were inverted, resulting in anatomical "hallucinations" (Figure 1). This issue stems from the distribution matching losses in the objective function that introduces/removes features from the image (Cohen et al., 2018; Zuo et al., 2023). In the absence of additional anatomical constraints, quantitative imaging beyond the lung region, focused on body composition assessment and radiomic feature assessment becomes difficult.

In this work, we hypothesize that skeletal muscle and SAT hallucinations can be mitigated by leveraging pre-computed TotalSegmentator (Wasserthal et al., 2023) labels in the multipath cycleGAN model. Specifically, we replace the identity loss with a tissue based loss function that penalizes intensity shifts in anatomical structures to preserve anatomy. We compare our approach against Stage one of the multipath cycleGAN model without anatomical guidance, the standard cycleGAN (Zhu et al., 2017) and switchable cycleGAN (Yang et al., 2021) on the following tasks: a) emphysema on paired kernels b) emphysema on

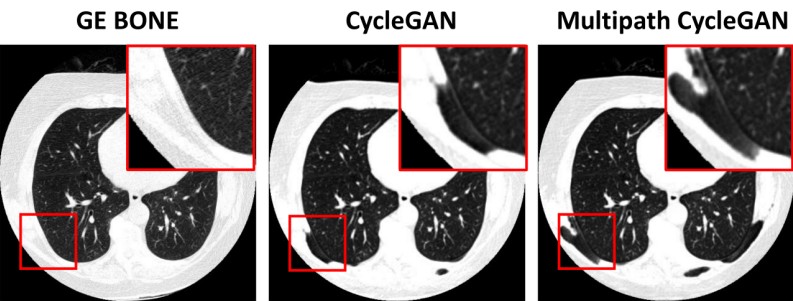

Figure 1: Distribution matching losses in cycleGAN and its variants result in anatomical hallucinations. Harmonization of the GE BONE image to the reference Siemens B30f kernel enforces consistent texture across the lung. However, we observe that in the highlighted regions and beyond, skeletal muscle and SAT tissue intensities are inverted, leading to inconsistent anatomy.

unpaired kernels and c) quantifying hallucinations in skeletal muscle and SAT post harmonization using Dice coefficients and effect sizes.

## 2. Methodology

A standard cycleGAN model uses two generators and discriminators for unpaired image-to-image translation. In the multipath cycleGAN model, we split a ResNet generator (He et al., 2016) into an encoder, shared residual block and decoder such that each domain has an encoder and decoder pair that harmonizes through the shared residual block, behaving as the latent space. Additionally, each domain has a PatchGAN discriminator (Isola et al., 2017) that promotes adversarial training by classifying $70 \times 70$ patches of images as real or synthetic. In Stage one of the multipath cycleGAN model, we use reconstruction kernels from the Siemens B50f(hard), Siemens B30f(soft), GE BONE (hard) and GE STANDARD (soft) kernels from the National Lung Cancer Screening Trial (NLST) (Aberle et al., 2011) dataset, harmonizing across six different combinations of reconstruction kernels. For each path, the cycleGAN operates as follows: in the forward path, the source encoder maps the real input to the shared latent space, and the target decoder produces a synthetic image in the target domain style. In the backward path, the target encoder maps the real image to the latent space, and the source decoder generates the synthetic image in the source domain style. The PatchGAN discriminator distinguishes between real and synthetic images to promote adversarial learning.

Previously, we introduced an identity loss in addition to the adversarial and cycle-consistency losses to preserve the radio-opacity in the harmonized images. This identity loss, computed as the mean squared error between the downsampled versions of the real and synthetic images, smooths out kernel effects and encourages the model to preserve intensity across the entire image. However, the identity loss primarily focuses on preserving global structural consistency, which may overlook subtle intensity variations in localized structures.

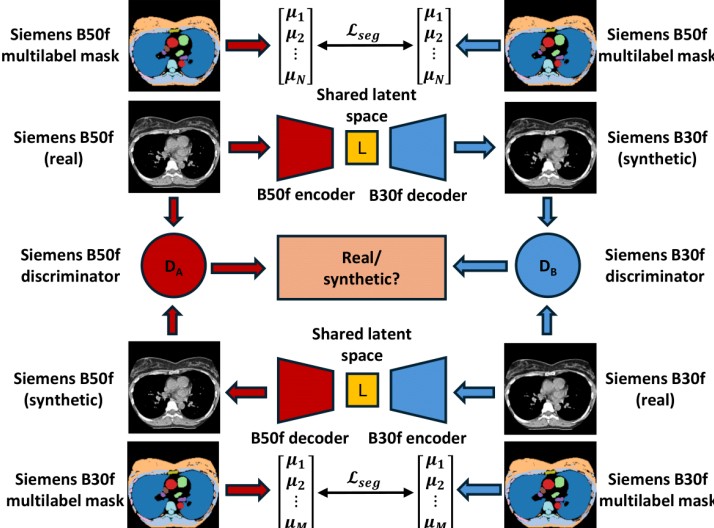

Figure 2: For any given pair of reconstruction kernels, the generator is a ResNet, formed by a source encoder and target decoder in the forward path and a target encoder and source decoder in the backward path. Each generator produces a synthetic image with the style of either the source or target kernel. A PatchGAN is used as a discriminator for the corresponding domain to distinguish between real and synthetic images. The mean of all unique labels are computed using the multilabel masks for the real and synthetic image and are penalized such that the anatomy remains preserved in the harmonized image.

To address this, we replace the identity loss with a tissue statistic loss that penalizes the mean intensity difference for each anatomical structure between the real and synthetic images. The tissue statistic loss uses anatomical labels derived from TotalSegmentator (version 2.1.0). The mean intensity is chosen as the statistic with the assumption that mean intensities between hard and soft kernels are similar. As shown in Fig. 2, our proposed approach follows the multipath cycleGAN architecture with the tissue statistic loss replacing the identity loss to preserve anatomy.

We train our proposed multipath cycleGAN model with anatomical guidance using 100 volumes per kernel, reconstructed with Siemens B50f, Siemens B30f, GE BONE and GE STANDARD (soft). We train on axial image slices and multilabel masks of size $512 \times 512$ pixels, clipping intensities to [-1024, 3072] Hounsfield units (HU) and normalizing to [-1,1]. The model is trained for 200 epochs in parallel on two NVIDIA A6000 RTX GPUs with a batch size of four, Adam (Kingma, 2014) optimizer and a learning rate of $2 \times 10^{-3}$, which remains constant for 100 epochs before linearly decaying. The generator and discriminator are governed by an adversarial loss using the LSGAN (Mao et al., 2017) loss function. We implement the cycle-consistency loss following the standard cycleGAN model, weighted by a factor of 10. The tissue statistic loss, which is computed as the mean squared error

(MSE) between the mean intensities of anatomical labels in the real and synthetic images, is weighted by a factor of 0.01.

## 3. Experiments

### 3.1. Baseline models

We compare our proposed approach to three different baseline models: standard cycle-GAN (Zhu et al., 2017), switchable cycleGAN (Yang et al., 2021) and the multipath cycleGAN without anatomical guidance (Krishnan et al., 2025). We implement 4 individual standard cycleGAN models and switchable cycleGAN models for all paired kernels and unpaired kernels that harmonize to a reference soft kernel. We trained each standard cycleGAN for 200 epochs using a batch size of six, Adam optimizer for the generator and discriminator and learning rate of $2 \times 10^{-3}$ . Each switchable cycleGAN was trained for 200 epochs using a batch size of 16, patch size of $128 \times 128$ pixels, Adam optimizer and a learning rate of of $1 \times 10^{-5}$. The multipath cycleGAN model without anatomical guidance is trained with the same hardware and optimization settings as the anatomy guided multipath cycleGAN, with a batch size of two.

### 3.2. Evaluation using percent emphysema

Generative networks vary in performance during every epoch as a result of adversarial learning. We use 100 volumes from each representative kernel for validation that results in the checkpoint to be used during inference. We select the optimal model checkpoint for inference using percent emphysema as our validation metric. Using a publicly available algorithm (Hofmanninger et al., 2020), we identify lung regions and compute percent emphysema by quantifying the percentage of voxels that have radio-opacity less than -950 HU. We consider the following paths for checkpoint selection: Siemens B50f to Siemens B30f, GE BONE to Siemens B30f and GE STANDARD to Seimens B30f. MSE is computed for emphysema scores on paired kernels, while Kullback-Leibler (KL) divergence is used for unpaired kernels. We rank all the scores, performing a weighted sum of ranks to obtain the overall rank for the given epoch using the equation:

$$\text{Epoch} = 0.5 \cdot \text{MSE}_{\text{B50f} \to \text{B30f}} + 0.25 \cdot \text{KL}_{\text{BONE} \to \text{B30f}} + 0.25 \cdot \text{KL}_{\text{STANDARD} \to \text{B30f}} \quad (1)$$

The epoch with smallest rank is chosen as the optimal checkpoint for inference.

### 3.3. Quantifying tissue hallucinations using TotalSegmentator

We investigate the consistency in skeletal muscle and SAT post harmonization for the GE kernels. Using Dice coefficients, we quantify the magnitude of hallucinations on the unpaired kernels for every available model. Cohen's d (Cohen, 2013) is computed on the Dice scores obtained for muscle and SAT to quantify the effect size between our proposed approach and the baseline models. Cohen's d quantifies the magnitude of differences between two groups by considering the means and pooled standard deviation. Effect sizes to quantify hallucinations are categorized as small (d≤0.2), medium (0.2≤d≤0.5), and large (d≥0.8). A positive d indicates a higher mean in group one compared to group two, while a negative d indicates the opposite.

Table 1: Percent emphysema measurements before and after harmonization across all models for paired kernels from the Siemens and GE manufacturers. Measurements are represented as RMSE and 95% confidence intervals. Differences before and after harmonization are significant ($p < 0.05$, paired Wilcoxon signed-rank test). $^*p < 0.05$ compared to other methods.

|  | Siemens | GE |
|---|---|---|
| Before harmonization | 12.12% [11.76%, 12.42%] | 9.91% [9.5%, 10.33%] |
| CycleGAN | 1.16% [1.05%, 1.29%] | 0.91% [0.81%, 1.05%] |
| Switchable CycleGAN | 1.57% [1.41%, 1.71%] | 1.01% [0.89%, 1.21%] |
| Proposed (w/o guidance) | 1.35% [1.22%, 1.52%] | **0.84% [0.75%, 0.95%]**$^*$ |
| Proposed (w/ guidance) | **1% [0.9%, 1.11%]**$^*$ | 1.05% [0.94%, 1.25%] |

## 4. Results

We compare performance of the proposed approach with the baseline models on emphysema quantification and anatomical consistency using 240 withheld volumes.

### 4.1. Emphysema quantification on paired and unpaired kernels

We investigate the impact of kernel harmonization on paired reconstruction kernels obtained from Siemens and GE manufacturers. We use bootstrapping with 1000 resamples to compute the median (RMSE) and 95% confidence intervals. All before and after measurements are significant ($p < 0.05$, Wilcoxon-signed rank test). All measurements are presented in Table 1. Before harmonization, there is variability in the emphysema measurements for the Siemens (12.12% [11.76%, 12.42%]) and GE (9.91% [9.5%, 10.33%]) kernels. Harmonization minimizes differences in emphysema measurements across all models with the anatomy-guided multipath cycleGAN achieving the lowest RMSE for Siemens (1%[0.9%,1.11%]) and the counterpart model without anatomical guidance achieving the lowest RMSE for GE (0.84% [0.75%, 0.95%]).

Across reconstruction kernels from all manufacturers, before harmonization, the B50f and BONE kernels have median emphysema scores of 20.02% and 12.75% while the B30f (reference) and STANDARD kernels have scores of 6.60% and 2.34%. We present the minimum, maximum, and median emphysema scores in Table 2. Compared to the counterpart model without guidance, anatomical guidance improved consistency in emphysema measurements with a small improvement in the B50f kernel, while median emphysema scores for BONE and STANDARD were slightly lower. Across other baselines, the proposed model with anatomical guidance showed consistent emphysema measurements.

### 4.2. Quantifying tissue hallucinations in unpaired kernel harmonization

The baseline models show consistent lung texture for the GE kernels when they are harmonized to the reference B30 kernel. However, hallucinations are introduced in specific regions of the skeletal muscle and SAT tissues where the intensities are labeled as lung (Figure 3). For the images shown in Fig 3, the proposed anatomical loss maintains the tissue intensity, effectively preserving the underlying anatomy. The anatomical guidance

Table 2: Percent emphysema measurements for all kernels harmonized to the reference Siemens B30f with a median emphysema measurement of 6.60% (0.22%, 40.54 %) kernel across all models. All measures are expressed as median (minimum, maximum). Mann-Whitney U test is used to assess statistical significance before and after harmonization

|  | Siemens B50f | GE BONE | GE STANDARD |
|---|---|---|---|
| Before harmonization | 20.02% (2.32%, 43.08%) | 12.75% (0.75%, 43.59%) | 2.34% (0.02%, 42.38%) |
| CycleGAN | 7.19% (0.23%, 40.78%) | **6.00% (0.38%, 50.07%)** | 5.56% (0.30%, 59.43%) |
| Switchable CycleGAN | 7.30.% (0.21%, 42.36%) | 10.01% (1.28%, 55.54%) | **6.53% (0.15%, 56.37%)** |
| Proposed (w/o guidance) | 6.34% (0.27%, 38.92%) | 5.70% (0.11%, 50.24%) | 6.01% (0.22%, 53.13%) |
| Proposed (w/ guidance) | **6.73% (0.24%, 40.00%)** | 5.11% (0.14%, 46.03%) | 5.25% (0.16%, 45.77%) |

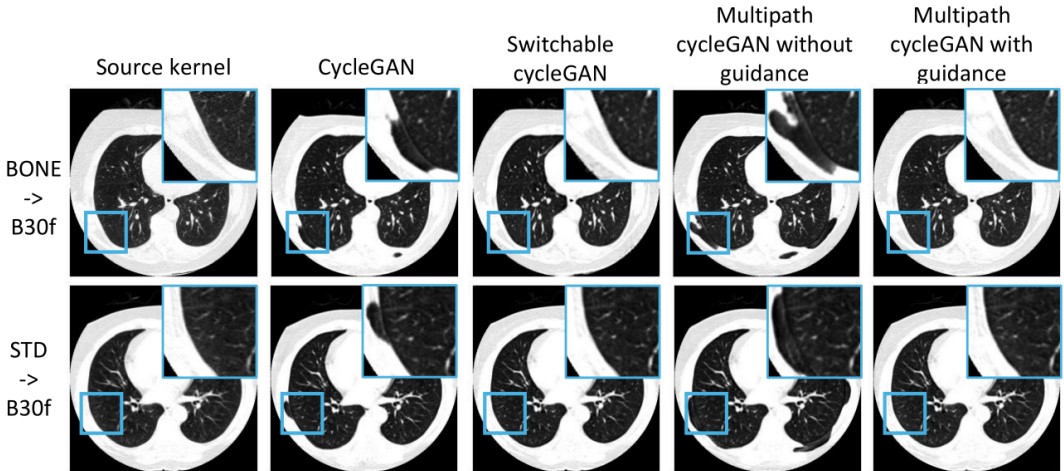

Figure 3: Unpaired kernel harmonization using distribution matching losses enforces consistency in the lung. However, in the cycleGAN and multipath cycleGAN models, skeletal muscle and SAT tissue intensities are inverted, resulting in anatomical hallucinations. Using the anatomical guidance as a regularization constraint, we observe that hallucinations in the regions of interest disappear.

Table 3: Cohen's d to quantify anatomical hallucinations in skeletal muscle and SAT. Effect sizes to quantify hallucinations are categorized as small ($d \leq 0.2$), medium ($0.2 \leq d \leq 0.5$), and large ($d \geq 0.8$). Numbers are calculated between proposed with anatomy guide "`Proposed w/ guide`" and other comparison methods.

| Proposed w/ guide | CycleGAN | | Proposed w/o guide | | Switch. CycleGAN | |
|---|---|---|---|---|---|---|
|  | Muscle | SAT | Muscle | SAT | Muscle | SAT |
| **BONE → B30f** | 2.05 | 1.77 | 0.47 | 1.01 | 0.34 | −0.07 |
| **STD. → B30f** | 6.19 | 2.45 | 1.69 | 1.39 | 0.27 | −0.29 |

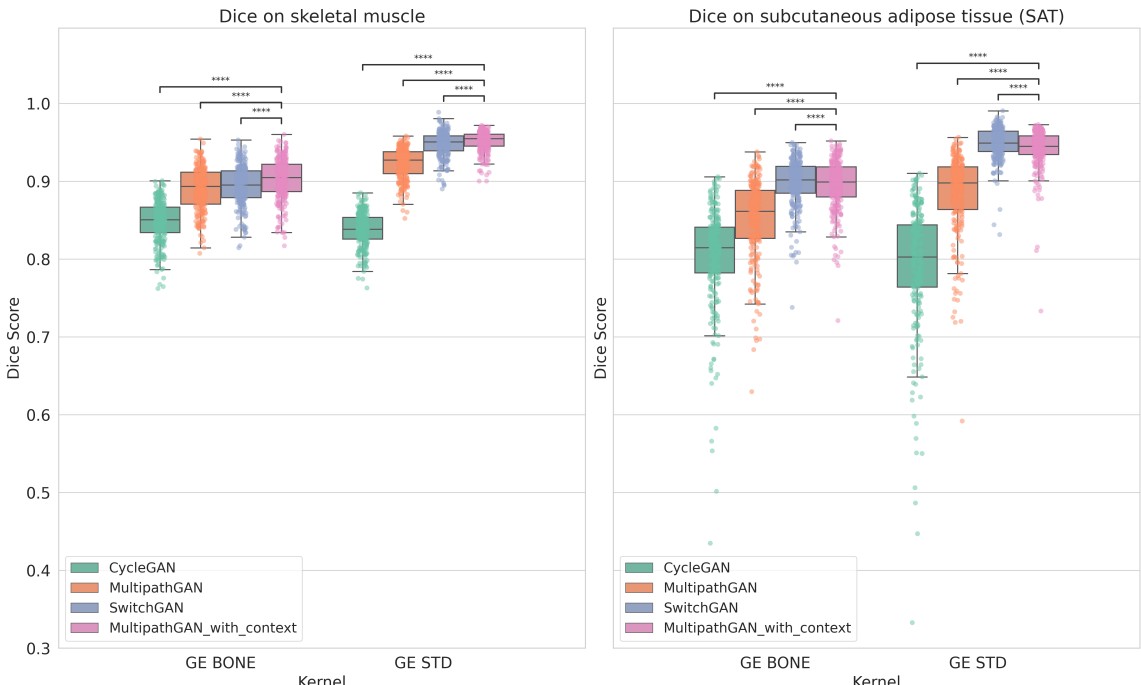

Figure 4: In the GE kernel, Dice scores computed on skeletal muscle and SAT highlight the degree of hallucinations. Anatomical guidance improves Dice scores on muscle and SAT compared to our counterpart model and the cycleGAN model. While Dice scores are better on skeletal muscle, the switchable cycleGAN shows slightly superior performance.

improved the median Dice scores to 0.90 for GE BONE and 0.95 for GE STD on skeletal muscle. Similarly, on SAT, median Dice improved to 0.90 for GE BONE and 0.94 for GE STD (Figure 4). However, performance on Dice was variable when compared to the switchable cycleGAN. We present all Cohen's d statistic values that determine effect size in Table 3. For the BONE kernel, effect sizes were large for skeletal muscle and SAT on cycleGAN, and medium for muscle and large for SAT on the multipath cycleGAN. For the STANDARD kernel, effect sizes were large for both models. Effect sizes ranged from small to medium for the switchable cycleGAN model across both kernels.

### 4.3. Performance of proposed approach on external dataset

We evaluate our proposed model on the NoduleVU dataset, a newly developed multimodal dataset from Vanderbilt University Medical Center. This dataset includes imaging and electronic health records acquired under Vanderbilt IRB #140274. For our study, we select ten scans reconstructed from the Siemens B50f, Siemens B30f, GE BONE, and GE STANDARD kernels after quality assurance. These scans are completely unpaired, meaning no ground truth exists between or across vendors. We assess our model's performance in emphysema quantification and muscle/SAT assessment after harmonizing all kernels to

match the style of the Siemens B30f kernel.The B30f kernel images had a distribution of 0.39% (0.01%, 10.11%). Before harmonization, the emphysema distributions for B50f, BONE and STANDARD images were 3.84% (1.00%, 22.44%), 3.37% (0.26%,22.83%) and 1.86% (0.01%, 8.66%). Post harmonization, the emphysema distributions for B50f, BONE and STANDARD were 1.12% (0.60%, 5.33%), 2.10% (0.23%, 18.32%) and 3.56% (0.25%, 14.47%). The B50f and BONE kernel images showed minimization in emphysema scores while the STANDARD kernel images showed a larger range of scores after harmonization. For the assessment of skeletal muscle and SAT, the BONE kernel and STANDARD kernel showed median Dice scores of 0.93, 0.95 for skeletal muscle and 0.91, 0.95 for SAT.

## 5. Discussion

In this work, we propose a tissue statistic loss in the multipath cycleGAN model that serves as anatomical guidance. The tissue statistic loss is implemented by penalizing mean intensity differences in local anatomical structures, guiding the model to preserve anatomical structures. While our proposed approach obtained consistent measurements for emphysema on the unpaired GE BONE and STANDARD kernels, the scores compared to the counterpart model were slightly lower. We believe that since the proposed loss function penalizes deviations from the mean, it may smooth out extreme intensity values beyond the threshold for emphysema quantification, reducing the number of detected emphysematous voxels. Future work could address this by incorporating robust higher-order statistical loss functions. On the external dataset, our approach minimized emphysema scores in the images reconstructed with the Siemens B50f and GE BONE kernels while the STANDARD kernel images showed slightly larger emphysema scores. The large median Dice scores on the external dataset highlight the preservation of anatomical structures post harmonization. Although our model leverages all possible anatomical labels from TotalSegmentator, the generalizability to other clinical tasks and organs requires further exploration. By leveraging precomputed anatomical labels from TotalSegmentator, our proposed method achieves higher Dice scores and shows large effect sizes on muscle and SAT compared to our counterpart model, demonstrating that the tissue statistic loss improved anatomical consistency post harmonization.

## Acknowledgments

This research was funded by the National Cancer Institute (NCI) grant R01 CA253923-04, R01 CA 253923-04S1. This work was also supported in part by the Integrated Training in Engineering and Diabetes grant number T32 DK101003. This research is also supported by the following awards: National Science Foundation CAREER 1452485; NCI U01 CA196405; UL1 RR024975-01 of the National Center for Research Resources and UL1 TR000445-06 of the National Center for Advancing Translational Sciences; Martineau Innovation Fund grant through the Vanderbilt-Ingram Cancer Center Thoracic Working Group; NCI Early Detection Research Network grant 2U01CA152662. The Vanderbilt Institute for Clinical and Translational Research (VICTR) is funded by the National Center for Advancing Translation Science Award (NCATS), Clinical Translational Science Award (CTSA) Program, Award Number 5UL1TR002243-03. The content is solely the responsibility of the authors and does not necessarily represent the official views of the NIH. We use generative AI to create code segments based on task descriptions, as well as debug, edit, and auto-complete code. Additionally, generative AI technologies have been employed to assist in structuring sentences and performing grammatical checks. It is imperative to highlight that the conceptualization, ideation, and all prompts provided to the AI originate entirely from the authors' creative and intellectual efforts. We take accountability for the review of all content generated by AI in this work.

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

## Appendix A. Multipath cycleGAN architecture

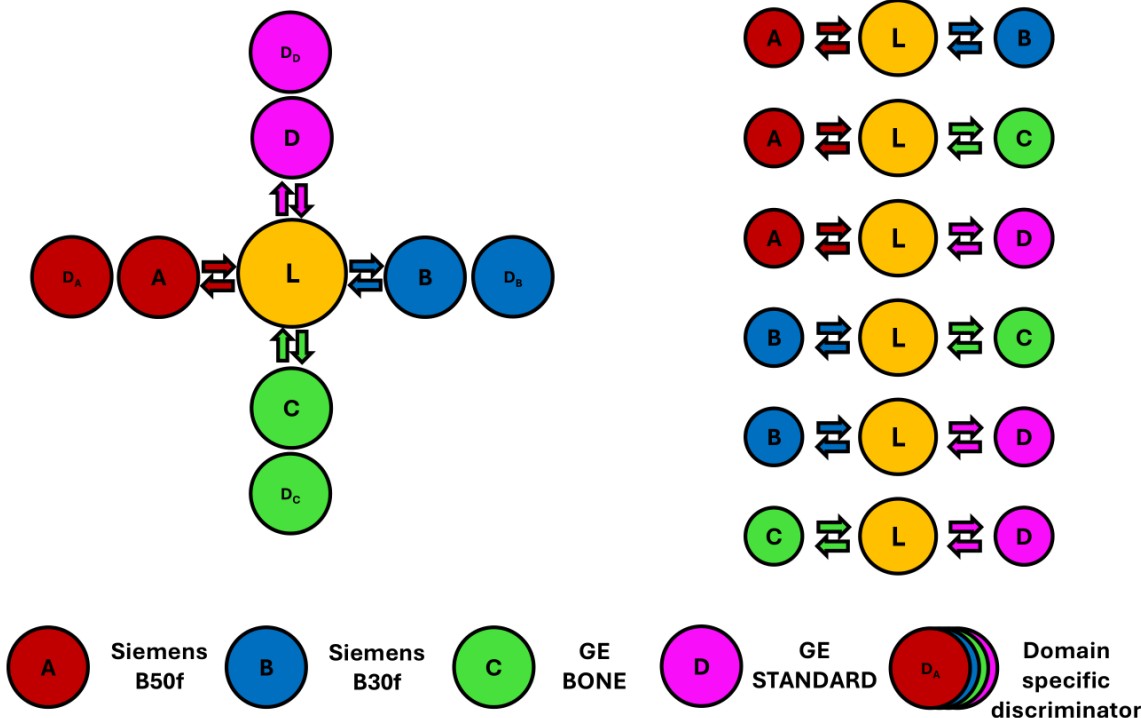

Figure 5: Multi-domain kernel harmonization between paired and unpaired reconstruction kernels can be accomplished through a multipath cycleGAN model housing a shared latent space, domain specific encoders, decoders and discriminators. For a given path, the source encoder maps the input to the latent space that is translated to an image in the style of the target domain by the target decoder.

## Appendix B. Inspection of visual quality on harmonized images

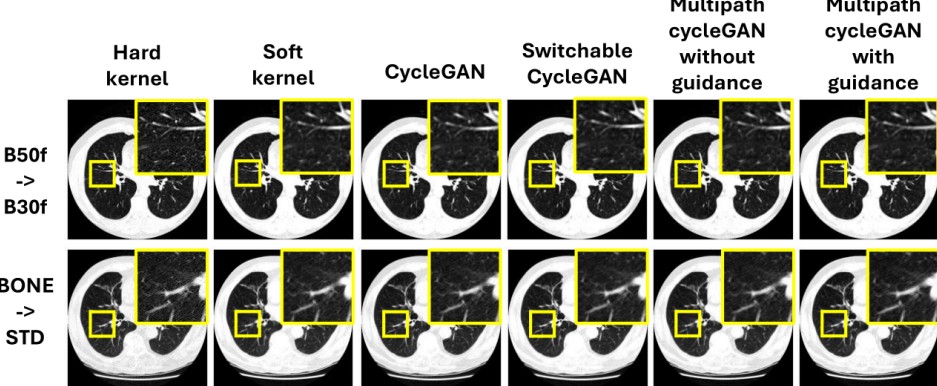

Figure 6: In paired reconstruction kernels, there exists a one-to-one pixel correspondence between the hard and soft kernel for a given vendor with differences in pixel noise. The hard kernel sharpens the lung parenchyma while the soft kernel smoothens it. Harmonization to the corresponding soft kernel image enforces consistent texture in the lung parenchyma across the baseline models and the proposed multipath cycleGAN model with anatomical guidance.

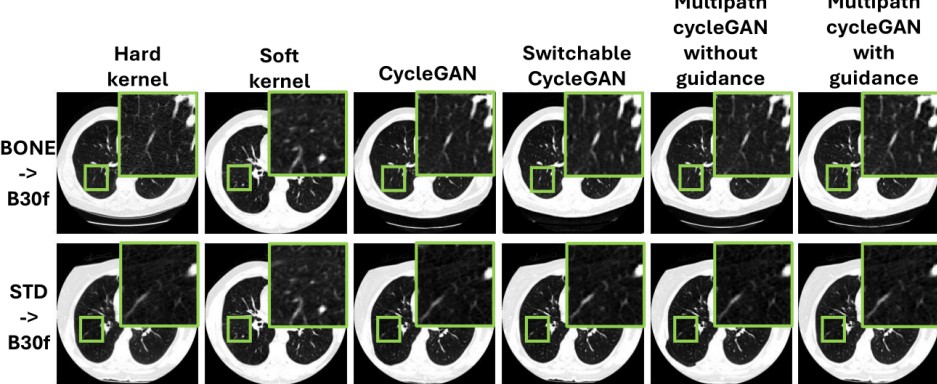

Figure 7: Unpaired reconstruction kernels exhibit differences in the positional alignment of subjects in addition to variations in reconstruction protocol. Harmonization to a reference Siemens B30f soft kernel enforces consistent texture in the lung parenchyma that benefits emphysema quantification. While some of the baseline models show tissue hallucinations in the muscle and fat regions, the proposed multipath cycleGAN model with anatomical guidance and the switchable cycleGAN model preserve the structure of the muscle and fat regions.

## Appendix C.  Quality of emphysema maps on harmonized images

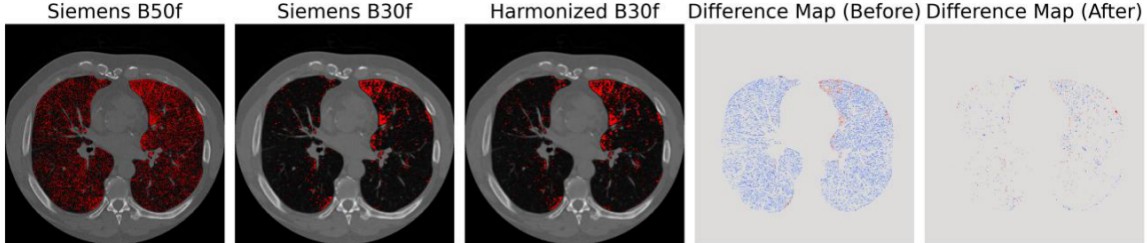

Figure 8:  Hard kernels overestimate emphysema as compared to soft kernels.  The PSNR and SSIM between the hard and soft kernel images were 35.72 dB and 0.89.  Harmonization improves the PSNR and SSIM to 48.16 dB and 0.99, indicating that image quality improved post harmonization.  Harmonization minimizes differences in emphysema measurements, enforcing similar emphysema patterns between the soft kernel and harmonized image.

## Appendix D.  Analyzing failure modes

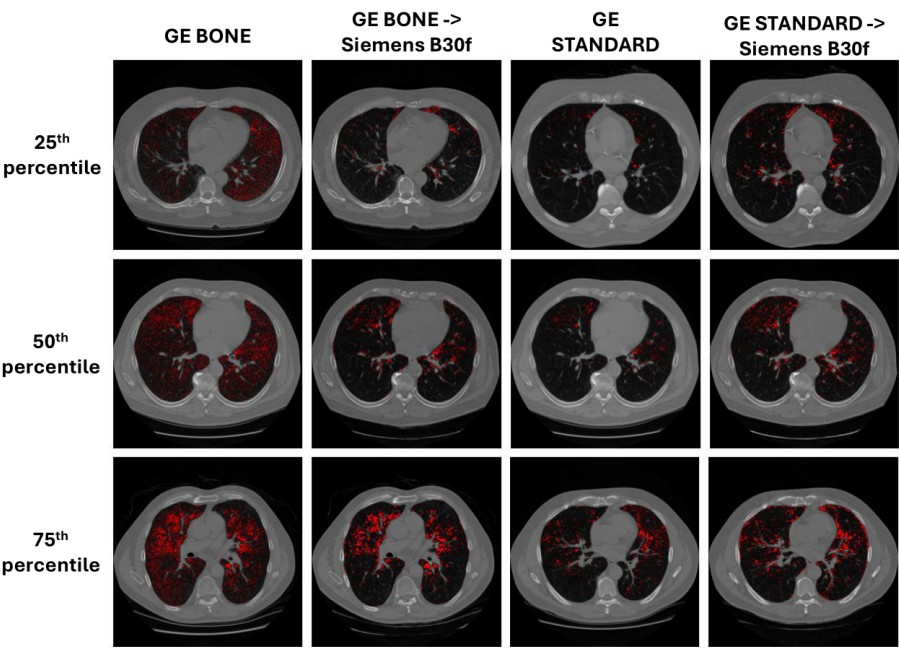

Figure 9:  Reconstruction kernels show variability in emphysema scores.  Harmonization of unpaired kernels to the style of the reference enforces consistency in emphysema score.  However, consistency in emphysema is variable across different subjects.

## Appendix E. Computational overhead of proposed approach

We train our multipath cycleGAN model with anatomical constraints using axial slices and the corresponding multilabel mask of size $512 \times 512$ pixels. We train our model for 200 epochs using a batch size of four, the Adam optimizer, a learning rate of $2 \times 10^{-3}$ and a linear learning rate scheduler where the learning rate remains constant for the first 100 epochs and decays linearly for the remaining 100 epochs. We train this model on two NVIDIA RTX A6000 GPUs (48 GB memory each). Our model consists of 56.156 million trainable parameters compared to 28.258 million for a standard cycleGAN and 40.132 million for a switchable cycleGAN. Given the number of trainable parameters and complex model architecture, training on large datasets consisting of 16343 images for the Siemens kernel and 14614 images for the GE kernels would require two weeks. We speed up training by randomly sampling 20% of the entire training dataset in each epoch such that our model covers all data samples throughout training. This approach reduced the training time to approximately 92 hours. We stabilize model training by leveraging PyTorch mixed precision to minimize memory consumption. During inference, our model harmonizes a given image within seconds on an NVIDIA RTX A6000 GPU using the corresponding encoder-decoder models. While our approach has a large computation overhead, our model enables training across diverse datasets through a shared latent space, facilitating multi-domain kernel harmonization.

