# OpenReview forum: "Anatomy-Guided Multi-Path CycleGAN for Lung CT Kernel Harmonization"
_MIDL.io/2025/Conference — MIDL 2025 Oral_

### Official Review · Reviewer_P9sA · 2025-02-22

**Confidence:** 4
**Preliminary Rating:** 4
**Recommendation:** Oral, Poster

**Summary:**

The paper proposes a novel anatomically guided multi-path CycleGAN framework for CT kernel harmonization. Specifically, the authors embed a “tissue statistic loss” into a multi-path CycleGAN architecture, enabling the model to penalize mean intensity differences in labeled anatomical regions, thereby preserving structure outside the lung during kernel harmonization. Experiments use large-scale data from different scanner manufacturers and kernels, showing that the proposed approach offers improved consistency in emphysema measurement and mitigates the anatomic hallucinations commonly observed in purely adversarial translation models. These findings suggest significant implications for more robust cross-center or longitudinal CT studies, fostering better reproducibility in quantitative imaging tasks.

**Strengths:**

1. The paper tackles an important clinical and technical challenge: reducing variability caused by heterogeneous CT reconstruction kernels across manufacturers.
2. The proposed “tissue statistic loss” is a methodological innovation, as it anchors the translation in explicit anatomical regions rather than relying solely on pixel-level adversarial or cycle-consistency objectives.
3. Experimental evidence is strong, with multiple kernels (soft and hard) from different vendors. This scope underscores the robustness and generalizability of the proposed method.
4. The authors provide their code, supporting reproducibility and further community development.
5. The inclusion of established segmentation tools (e.g. TotalSegmentator) for obtaining anatomical constraints demonstrates a thoughtful integration of existing resources, enhancing the method’s practical significance.

**Weaknesses:**

1. Although the tissue statistic loss significantly improves anatomical preservation, the method’s performance on certain global metrics can be slightly lower than the non-guided multipath CycleGAN. The trade-off between anatomical fidelity and some quantitative metrics might warrant further exploration.
2. The paper focuses primarily on lung emphysema quantification and muscle/SAT evaluation. Other clinical metrics or more diverse radiomic features are only briefly mentioned.
3. The computation of mean intensities within labeled regions is a simplistic statistic. Future works might explore robust statistics or more advanced similarity measures for improved guidance.
4. Although the code is publicly released, details about the computational overhead introduced by additional segmentation steps and how it might impact large-scale clinical workflows are not thoroughly discussed.

**Detailed Comments:**

Please refer to the comments in weaknesses section

**Justification Of The Preliminary Rating:**

The paper addresses a critical problem in multicenter CT imaging studies—kernel inconsistencies—and provides a solution using a multi-path CycleGAN enhanced with an anatomical constraint. This addition is conceptually novel, leading to improved tissue preservation and consistent emphysema measurements. Although some minor concerns remain about the trade-off between anatomical preservation and certain quantitative metrics, and the scope of validation is somewhat focused, the overall contribution is strong. The method is well-executed, the experiments are comprehensive, and the code release promises broad impact.

**Questions To Address In The Rebuttal:**

1. Could the authors elaborate on the observed slight decrease in emphysema measurement performance for some comparisons when using the tissue statistic loss? Are there plans to balance or tune the anatomical preservation vs. quantitative metric performance?
2. How sensitive is the model to inaccuracies in the body composition labels? Are there any baseline experiments on deliberately noisy labels to test resilience?
3. Please provide approximate training times and memory requirements, especially given the additional segmentation-based constraints.
4. Could the authors clarify whether the approach might generalize to other tasks or organs?

---

> ### Author Response · Authors · 2025-03-08
> **Response to reviewer P9sA**
>
> We thank the reviewer for their comprehensive summary and valuable feedback on our work. We address the weaknesses and rebuttal questions in detail:
> 1) **Tradeoff between anatomical fidelity and quantitative metrics**: We appreciate the comment regarding trade-off between anatomical fidelity and global performance metrics. In this work, our primary focus was to mitigate hallucinations generated by our counterpart model. We agree that this trade-off could impact global metrics which requires further exploration in future works.
> 2) **Other clinical tasks and radiomic features**: We agree that other clinical tasks and radiomic features were briefly mentioned. Our study primarily focuses on comparing our proposed approach with our counterpart model on emphysema quantification and muscle/SAT hallucination. Investigation of other downstream tasks and radiomic features is an important direction for future work.
> 3) **Simple mean statistic for the loss function**: We agree that the mean is a simple statistic and plan to explore robust statistics and other image similarity features in the future.
> 4) **Computational overhead and significance in clinical workflows**:  Our model leverages precomputed segmentation labels during training to penalize the mean intensities in each structure to preserve anatomical structure. During inference, the model can be applied to diverse datasets in clinical workflow without the need for segmentation labels. We explain the computational overhead of our model as follows: **We train our multipath cycleGAN model with anatomical constraints using axial slices and the corresponding multilabel mask of size 512 × 512 pixels. We train our model for 200 epochs using a batch size of four, the Adam optimizer, a learning rate of 2 × $10^{-3}$ and a linear learning rate scheduler where the learning rate remains constant for the first 100 epochs and decays linearly for the remaining 100 epochs. We train this model on two NVIDIA RTX A6000 GPUs (48 GB memory each). Our model consists of 56.156 million trainable parameters compared to 28.258 million for a standard cycleGAN and 40.132 million for a switchable cycleGAN. Given the number of trainable parameters and complex model architecture, training on large datasets consisting of 16343 images for the Siemens kernel and 14614 images for the GE kernels would require two weeks. We speed up training by randomly sampling 20% of the entire training dataset in each epoch such that our model covers all data samples throughout training. This approach reduced the training time to approximately 92 hours. We stabilize model training by leveraging PyTorch mixed precision to minimize memory consumption. During inference, our model harmonizes a given image within seconds on an NVIDIA RTX A6000 GPU using the corresponding encoder-decoder models. While our approach has a large computation overhead, our model enables training across diverse datasets through a shared latent space, facilitating multidomain kernel harmonization.** We have included the highlighted paragraph in **Appendix E** in our manuscript.
> 5) **Slight decrease in emphysema measurement**: We appreciate this insightful question from the reviewer. The tissue statistic loss penalizes differences in mean intensities between real and synthetic images for a given path in the model. Emphysema is quantified as the percentage of lung voxels with a radio-opacity below –950 Hounsfield units. Since the tissue statistic loss penalizes deviations from the mean, it may smooth out extreme intensity values beyond this threshold, reducing the number of detected emphysematous voxels. Future work could address this trade-off by incorporating robust higher-order statistical loss functions.
> 6) **Sensitivity of model to noisy labels**: We appreciate the reviewer’s suggestion. While we did not conduct experiments with noisy labels, as it falls beyond the scope of this work, we recognize the value of such an analysis. Future work could explore this sensitivity in more detail.
> 7) **Generalizability to other tasks and organs**: We thank the reviewer for this question. We leverage all possible anatomical labels from TotalSegmentator during training. Our assessment was focused on emphysema measurements in the lung and structural consistency in muscle and SAT compared to our counterpart model. The generalizability of our model to other tasks and organs remains an open question and warrants further exploration.

---

> ### Comment · Area_Chair_eCtC · 2025-03-14
> **Please provide final rating based on the authors' rebuttal**
>
> Please provide final rating based on the authors' rebuttal. Thank you!

---

### Official Review · Reviewer_Mmti · 2025-02-22

**Confidence:** 5
**Preliminary Rating:** 5
**Recommendation:** Oral

**Summary:**

1. This paper proposes an anatomy-guided multi-path CycleGAN that harmonizes CT reconstruction kernels by integrating anatomical information via a novel tissue statistic loss.
2. The method leverages pre-computed anatomical labels from TotalSegmentator to penalize mean intensity differences in key tissues.
3. Trained and evaluated on the NLST dataset, the approach shows superior performance in both emphysema quantification and the preservation of skeletal muscle and subcutaneous adipose tissue compared to standard CycleGAN variants.

Overall, the method effectively reduces measurement variability across different manufacturers while mitigating anatomical hallucinations.

**Strengths:**

1. The overall methodology is innovative. The integration of anatomical guidance via a tissue statistic loss into a multi-path CycleGAN framework is a novel contribution that addresses common artifacts seen in kernel harmonization.
2. The paper presents comprehensive experiments comparing standard CycleGAN, switchable CycleGAN, and a baseline multipath CycleGAN, demonstrating improvements in both quantitative metrics and qualitative anatomical consistency.
3. By preserving anatomical structures during harmonization, the approach improves the reliability of quantitative CT measurements, which is crucial for multi-center studies and longitudinal analyses.

**Weaknesses:**

1. The study uses 100 scans per kernel, which may limit the generalizability of the findings across broader and more diverse datasets.
2. The paper could further discuss the associated computational overhead and training stability.

**Detailed Comments:**

1. The paper is well-organized and clearly explains the rationale behind replacing the identity loss with a tissue statistic loss.
2. The detailed experimental setup and statistical validation add robustness to the results.
3. Future work could explore ablation studies on the weight of the tissue statistic loss and extend the evaluation to additional datasets (or) vendors to further validate the method’s generalizability.

**Justification Of The Preliminary Rating:**

The paper introduces a novel anatomy-guided CycleGAN for CT harmonization, improving anatomical consistency and reducing hallucinations. Extensive experiments show robust, clinically impactful results. CT kernel harmonization is enhanced by integrating anatomical guidance into CycleGAN. A novel tissue loss cuts hallucinations, boosting both quantitative and qualitative imaging metrics.

**Questions To Address In The Rebuttal:**

1. How well does the proposed method generalize to datasets beyond the NLST, particularly with different scanner types or reconstruction protocols?
2. Can the authors provide further insights into the computational overhead introduced by the multi-path architecture with anatomical guidance?

---

> ### Author Response · Authors · 2025-03-08
> **Response to reviewer Mmti**
>
> We thank the reviewer for summarizing our work and providing valuable feedback. We address the weaknesses, detailed comments and rebuttal questions below:
> 1) **Number of training images**: We thank the reviewer for raising this question. Although we use 100 scans, we breakdown the volumes into their respective slices. This provides a large dataset of different slices across all the subjects. The images reconstructed with the Siemens kernels consisted of 16343 slices while the images reconstructed with the GE kernels had 14614 images.
> 2) **Generalizability to an external dataset beyond NLST**:We thank the reviewer for the question. We evaluated the performance of our model on an external test dataset. We describe the performance in **section 4.3** as follows:
> **We evaluate our proposed model on the NoduleVU dataset, a newly developed multimodal dataset from Vanderbilt University Medical Center. This dataset includes imaging and electronic health records acquired under Vanderbilt IRB #140274. For our study, we select ten scans reconstructed from the Siemens B50f, Siemens B30f, GE BONE, and GE STANDARD kernels after quality assurance. These scans are completely unpaired, meaning no ground truth exists between or across vendors. We assess our model’s performance in emphysema quantification and muscle/SAT assessment after harmonizing all kernels to match the style of the Siemens B30f kernel. The B30f kernel images had a distribution of 0.39% (0.01%, 10.11%). Before harmonization, the emphysema distributions for B50f, BONE and STANDARD images were 3.84% (1.00%, 22.44%), 3.37% (0.26%,22.83%) and 1.86% (0.01%, 8.66%). Post harmonization, the emphysema distributions for B50f, BONE and STANDARD were 1.12% (0.60%, 5.33%), 2.10% (0.23%, 18.32%) and 3.56% (0.25%, 14.47%). The B50f and BONE kernel images showed minimization in emphysema scores while the STANDARD kernel images showed a larger range of scores after harmonization. For the assessment of skeletal muscle and SAT, the BONE kernel and STANDARD kernel showed median Dice scores of 0.93, 0.95 for skeletal muscle and 0.91, 0.95 for SAT.**
> Our model effectively preserved the anatomical structure of muscle and SAT on the unseen external dataset. However, we observe that performance on emphysema quantifcation was variable. This could be attributed to the differences in population where the emphysema severity in the external dataset was different from the NLST dataset.
> 3) **Computational overhead**: We thank the reviewer for this question. We describe the computational overhead in **Appendix E** in the manuscript as follows: **We train our multipath cycleGAN model with anatomical constraints using axial slices and the corresponding multilabel mask of size 512 × 512 pixels. We train our model for 200 epochs using a batch size of four, the Adam optimizer, a learning rate of 2 × $10^{-3}$ and a linear learning rate scheduler where the learning rate remains constant for the first 100 epochs and decays linearly for the remaining 100 epochs. We train this model on two NVIDIA RTX A6000 GPUs (48 GB memory each). Our model consists of 56.156 million trainable parameters compared to 28.258 million for a standard cycleGAN and 40.132 million for a switchable cycleGAN. Given the number of trainable parameters and complex model architecture, training on large datasets consisting of 16343 images for the Siemens kernel and 14614 images for the GE kernels would require two weeks. We speed up training by randomly sampling 20% of the entire training dataset in each epoch such that our model covers all data samples throughout training. This approach reduced the training time to approximately 92 hours. We stabilize model training by leveraging PyTorch mixed precision to minimize memory consumption. During inference, our model harmonizes a given image within seconds on an NVIDIA RTX A6000 GPU using the corresponding encoder-decoder models. While our approach has a large computation overhead, our model enables training across diverse datasets through a shared latent space, facilitating multidomain kernel harmonization.**
> 4) **Ablation study**: We thank the reviewer for the suggestion. We acknowledge that conducting an ablation study on the weight of the tissue loss would provide valuable insights into its impact on model performance. This is an important direction for future work.

---

> ### Comment · Area_Chair_eCtC · 2025-03-14
> **Please provide final rating based on the authors' rebuttal**
>
> Please provide final rating based on the authors' rebuttal. Thank you!

---

### Official Review · Reviewer_vLgJ · 2025-02-26

**Confidence:** 4
**Preliminary Rating:** 2
**Recommendation:** Poster
**Final Rating:** 4

**Summary:**

In this work, the authors propose a multi-path cycleGAN framework for kernel harmonization. The framework incorporates multi-region anatomical labels and a tissue statistic loss as anatomical regularization to preserve structural integrity. The anatomical labels are generated by leveraging TotalSegmentator. The hypothesis of the authors is that skeletal muscle and SAT hallucinations can be mitigated by leveraging anatomical  labels in the multipath cycleGAN model. For the experiments, the authors used 100 scans from NLST for training, and tested on 240 withheld scans. The results show more consistent emphysema measurements.

**Strengths:**

- The fact that kernel differences lead to different emphysema measurements is a real issue in clinical practice. It is often solved by pushing for standardized acquisition protocols, but this paper presents a method where you can mitigate it using AI methodology.
- Clearly written.
- Publicly available code.

**Weaknesses:**

- Limited novelty. A lot of research has come out where GANs are used. This framework also heavily relies on the original cycleGAN implementation. However, the use of anatomical labels from TotalSegmentator, I have not seen before.
- No visual checking of produced images on the test set.
- No validation for other datasets.
- No evidence that the emphysema scores are still of good quality. They are more consistent, sure, but are they still of good quality?
- No analysis of failure modes.

**Detailed Comments:**

- What if totalsegmentator makes mistakes?
- In what circumstances does this method fail?
- Please add which version of TotalSegmentator was used for this work, there are multiple versions available.
- GitHub repo with code contains no license. This makes re-use of the code problematic. Please add a proper license, for example MIT license.
- Manuscript contains quite some typos, for example 'swithcable'. Please correct.

**Justification Of The Final Rating:**

I would like to thank the authors for the extensive rebuttal, and the additional work added in the Appendices - really nice!
I think this has substantially improved the paper and I am happy to upgrade my score based on the provided additional info and evidence.

**Justification Of The Preliminary Rating:**

I am not fully convinced that the produced images are of sufficient quality for emphysema quantification (which seems to be the main use case), and find no evidence in the paper for this. In addition, I think the novelty of this work is limited.

**Questions To Address In The Rebuttal:**

- Please discuss failure modes.
- Please address my comment whether the produced images are indeed of sufficient quality to do emphysema measurements. Can you provide evidence for this?

**Special Issue:**

No

---

> ### Author Response · Authors · 2025-03-08
> **Response to reviewer vLgJ**
>
> We thank the reviewer for summarizing our work and providing valuable feedback We address the weaknesses, detailed comments, and rebuttal questions below:
> 1) **Clarification on Novelty**: We thank the reviewer for their feedback. We agree with the reviewer that GAN models have been implemented extensively in research. The innovation of our approach focuses on integrating labels from TotalSegmentator where a tissue statistic loss is computed to guide the model in preserving anatomy. We include a supplementary figure in **Appendix A** that highlights our model architecture.
> 2) **Visual checking on test set to highlight images are of sufficient quality**: We thank the reviewer for highlighting this uncertainty. We present qualitative comparisons on paired and unpaired kernels for the baseline and proposed model. We include these figures in **Appendix B**.
> 3) **Validation on external dataset**: We thank the reviewer for addressing this uncertainty. We evaluated the performance of our model on an external test dataset. We describe the performance in **section 4.3** as follows:
> **We evaluate our proposed model on the NoduleVU dataset, a newly developed multimodal dataset from Vanderbilt University Medical Center. This dataset includes imaging and electronic health records acquired under Vanderbilt IRB #140274. For our study, we select ten scans reconstructed from the Siemens B50f, Siemens B30f, GE BONE, and GE STANDARD kernels after quality assurance. These scans are completely unpaired, meaning no ground truth exists between or across vendors. We assess our model’s performance in emphysema quantification and muscle/SAT assessment after harmonizing all kernels to match the style of the Siemens B30f kernel. The B30f kernel images had a distribution of 0.39% (0.01%, 10.11%). Before harmonization, the emphysema distributions for B50f, BONE and STANDARD images were 3.84% (1.00%, 22.44%), 3.37% (0.26%,22.83%) and 1.86% (0.01%, 8.66%). Post harmonization, the emphysema distributions for B50f, BONE and STANDARD were 1.12% (0.60%, 5.33%), 2.10% (0.23%, 18.32%) and 3.56% (0.25%, 14.47%). The B50f and BONE kernel images showed minimization in emphysema scores while the STANDARD kernel images showed a larger range of scores after harmonization. For the assessment of skeletal muscle and SAT, the BONE kernel and STANDARD kernel showed median Dice scores of 0.93, 0.95 for skeletal muscle and 0.91, 0.95 for SAT.**
> 4) **Quality of emphysema scores**: We acknowledge the concern raised by the reviewer. In CT imaging, hard kernels overestimate emphysema scores compared to soft kernels, which are preferred for emphysema quantification [1]. Existing literature evaluates the efficacy of harmonization on paired kernels [2]. Consistency in scores is a measure of the quality of emphysema. Additionally, the harmonized images displayed emphysema patterns that were similar to the soft kernel. We include a figure in **Appendix C** that shows the emphysema maps on the hard, soft and harmonized images. In unpaired kernels, consistency to the reference distribution highlights quality of emphysema. Our focus is on the technical aspects of harmonization rather than the biological validity of emphysema.
> 5) **Failure modes and circumstances where the model could fail**: We thank the reviewer for highlighting this. We present a figure in **Appendix D** that showcases how performance on unpaired kernels is variable across different patients. Furthermore, we discuss cases where our model can fail:
> >*As CT scanner technology advances, manufacturers continue to introduce newer reconstruction kernels. Since our model was trained on kernels from a prospective study, it may require additional fine-tuning to generalize effectively to newer kernels.\
> >*Our method has been trained and tested on a specific dataset from a given population. If applied to a dataset from a completely different patient population, the model might fail to generalize.\
> >*The method relies on the assumption that the mean intensities between the hard and soft kernel images are similar. The approach can fail in cases where this assumption is not true.
> 6) **TotalSegmentator mistakes**: We acknowledge that TotalSegmentator can make mistakes where certain structures can be incorrectly labeled. However, we are interested in the mean statistic of these structures that can help with anatomical guidance.
> 7) We have included the MIT license and have corrected the typos in the manuscript.
>
> References:\
> [1] Boedeker, K. L., McNitt-Gray, M. F., Rogers, S. R., Truong, D. A., Brown, M. S., Gjertson, D. W., & Goldin, J. G. (2004). Emphysema: Effect of reconstruction algorithm on CT imaging measures. Radiology \
> [2] Gallardo-Estrella, L., Lynch, D. A., Prokop, M., Stinson, D., Zach, J., Judy, P. F., van Ginneken, B., & van Rikxoort, E. M. (2016). Normalizing computed tomography data reconstructed with different filter kernels: effect on emphysema quantification. European Radiology

---

### Author Rebuttal · Authors · 2025-03-08

**Rebuttal:**

We appreciate the valuable feedback from the reviewers and have addressed their comments in the comments section and the manuscript. We highlight the additional information, figures added to the manuscript with an asterisk *.
Reviewer 1:
1) We clarify that the novelty of this work involves a tissue-based loss that integrates anatomical information from TotalSegmentator to penalize structural differences post harmonization.
2) *We include figures to showcase visual quality of images on the test set (Appendix B).
3) We perform external validation on an unseen test dataset and report the performance of our model.
4) *We include a figure to demonstrate the quality of emphysema masks (Appendix C).
5) *We present a figure that addresses failure modes in unpaired kernels through variability in performance (Appendix D).
6) We clarify that TotalSegmentator can make mistakes. However, we are interested in the mean statistic of these structures that can help
  with anatomical guidance.
7) We have included the TotalSegmentator version (2.1.0) in our paper.
8) We have included an MIT license in our Github repository and corrected the typos in the manuscript.

Reviewer 2:
1) *We analyse the performance of our proposed approach on an external test dataset.
2) We address the number of scans used for training and clarify that each slice is an individual image that results in a large dataset.
3) We acknowledge the ablation study on the weight of the statistic loss as future work.
4) *We discuss the computational overhead of our proposed approach (Appendix E).

Reviewer 3:
1) We acknowledge that the trade-off between anatomical fidelity and quantitative metrics warrants further exploration, as it was beyond the scope of this work.
2) The scope of this paper was focused on emphysema and muscle/SAT assessment. We acknowledge that future work could explore other clinical tasks and radiomic features.
3) *We acknowledge that future work could explore robust statistics
4) *We address the decrease in emphysema performance due to the tissue loss.
5) We did not investigate noisy labels as it was beyond the scope of our work.
6) *We discuss the computational overhead of our model.
7) We acknowledge that generalizability to other tasks and organs requires further exploration.

We thank the reviewers again for their insightful feedback. We have addressed the concerns to the best of our abilities.
Kind regards,
Aravind Krishnan & co-authors

**Supporting Material:**

/attachment/e012d227a2a044902927fb6618bf97141ab195db.pdf

---

### Meta-Review · Area_Chair_eCtC · 2025-03-19

**Recommendation:** Accept (Poster)
**Confidence:** 5

**Metareview:**

This manuscript investigates computed tomography (CT) harmonization using a multi-path cycleGAN framework incorporating multi-region anatomical labels and a tissue statistic loss to preserve structural integrity. The work has practical significance given the heterogeneous appearance of CT scans due to acquisition and reconstruction differences. The study is written clearly and includes incremental improvements to reduce anatomic hallucinations. The evaluation is limited to a specific cohort (NLST) but rigorous. All reviewers found the proposed work worthy of acceptance.